# Silk Peptide Ameliorates Sarcopenia through the Regulation of Akt/mTOR/FoxO3a Signaling Pathways and the Inhibition of Low-Grade Chronic Inflammation in Aged Mice

**DOI:** 10.3390/cells12182257

**Published:** 2023-09-12

**Authors:** Hyun-Ji Oh, Heegu Jin, Jeong-Yong Lee, Boo-Yong Lee

**Affiliations:** 1Department of Food Science and Biotechnology, College of Life Science, CHA University, Seongnam 13488, Republic of Korea; guswl264@naver.com (H.-J.O.); heegu94@hanmail.net (H.J.); 2Worldway Co., Ltd., Sejong 30003, Republic of Korea; dalgoozi@hanmail.net

**Keywords:** acid-hydrolyzed silk peptide, sarcopenia, Akt, mTOR, FoxO3a, signaling pathway, aging, oxidative stress, inflammation

## Abstract

As populations around the world age, interest in healthy aging is growing. One of the first physical changes that occurs with aging is the loss of muscle mass and strength, termed sarcopenia. Sarcopenia limits the activity of older people, reduces their quality of life, and increases the likelihood of their developing disease. In the present study, we aimed to evaluate the effects of the ingestion of acid-hydrolyzed silk peptide (SP) on the muscle mass and strength of mice of >22 months of age with naturally occurring sarcopenia, and to identify the mechanisms involved. The daily administration of SP for 8 weeks increased the activation of the Akt/mTOR/FoxO3a signaling pathways and increased the muscle mass and strength of the old mice. In addition, SP inhibited oxidative stress and inflammation in muscle, which are direct causes of sarcopenia. Therefore, SP represents a promising potential treatment for sarcopenia that may improve the healthy lifespan and quality of life of older people.

## 1. Introduction

Given that societies around the world are aging, it is increasingly important to ensure the quality of life of older people and to prolong their healthy lifespan. Aging is a physiological process that increases the risks of various diseases, and one of the first physical changes that occurs with aging is a reduction in muscle mass and strength, termed sarcopenia. This is characterized by steady decreases in muscle mass and the number and size of muscle fibers [1]. These changes limit the range and quantity of activity that is possible, which leads to a deterioration in health, including a weakening of immunity [2]. For this reason, interventions targeting sarcopenia would permit the maintenance of a good quality of life and the prevention of disease in older people. However, although interest in sarcopenia is increasing, there are no specific methods of treating or preventing this condition at present. There are several existing strategies that improve muscle strength, but they involve exercises that are inappropriate for older people and individuals with acute muscle damage. Therefore, considering the potentially severe consequences of sarcopenia and the aging of societies, there is a considerable need for effective therapies for sarcopenia in older people.

Age-related sarcopenia commences when the rate of muscle protein breakdown exceeds the rate of synthesis [3]. Therefore, therapies aiming to slow the breakdown of muscle protein should be effective for sarcopenia. The forkhead box O (FoxO) subfamily of forkhead transcription factors is known to promote autophagy through several cellular mechanisms, thereby promoting cell survival, and FoxO3a specifically is recognized as a major regulator of muscle protein loss [4,5,6]. Continuous FoxO3a activity in muscle has been shown to increase the expression of the muscle-specific E3 ubiquitin ligases F-box protein (Fbx32, also known as atrogin) and muscle ring-finger 1 (MuRF1), leading to muscle protein degradation and a loss of muscle mass [7,8,9,10]. Therefore, the inhibition of FoxO3a activity may represent an effective means of reducing muscle wasting.

An alternative approach to the amelioration of sarcopenia would be to promote muscle protein synthesis. Insulin-like growth factor (IGF)-1 not only inhibits protein degradation by blocking the Akt/FoxO signaling pathway, but also stimulates protein synthesis through activation of the Akt/mechanistic target of rapamycin (mTOR) pathway [11,12]. mTOR is a key regulator of protein synthesis, and the inhibition of mTOR activity in muscle is closely associated with sarcopenia [13]. Therefore, activation of the Akt/mTOR pathway in muscle should ameliorate sarcopenia by promoting muscle protein synthesis. Furthermore, myogenesis is a process in which muscle satellite cells are assembled into contractile multinucleated muscle fibers [14]. The myogenic transcription factors myoblast determination protein 1 (MyoD), myocyte enhancer factor 2 (MEF-2), and myogenin play roles in inducing somatic cells to differentiate into myoblasts and in their differentiation into myotubes [15,16]. Thus, approaches that induce the expression of these factors may also delay sarcopenia.

Both inflammation and oxidative stress are also known to be involved in the pathogenesis of sarcopenia [17,18,19]. Immunosenescence, a gradual decline in immune function that occurs during the aging process, disrupts immune homeostasis and is associated with a low-grade chronic inflammatory state [20]. Pro-inflammatory signals derived from immune cells promote inflammation and induce muscle cell death [21,22]. Macrophages play an important role in skeletal muscle and are classified as M1 macrophages, which cause inflammation by secreting pro-inflammatory cytokines and M2 macrophages, which suppress excess inflammation and facilitate tissue repair [23]. Therefore, the balance between M1 and M2 macrophages is important for the prevention of muscle degradation. In addition, reactive oxygen species, which mediate many of the pathological features of oxidative stress in aging skeletal muscle, promote inflammation and sarcopenia [24]. Specifically, the accumulation of reactive oxygen species causes skeletal muscle atrophy, tissue degradation, and an impairment of muscle function [25,26]. Therefore, in the present study, we studied the effects of a candidate therapeutic substance on muscle inflammation and oxidative stress, which are key contributors to sarcopenia.

Acid-hydrolyzed silk peptide (SP), derived from *Bombyx mori* cocoons, is a material with various potential applications. Although there have been many previous studies of the potential therapeutic effects of SP, such as on immunosenescence and obesity [27,28,29], its effects on sarcopenia, a serious problem in older people, have not been studied in detail. SP principally comprises a low-molecular weight protein that can be quickly absorbed without the need for digestion, making it suitable for administration to older people with poor digestive function. Therefore, we aimed to determine the effect of SP on the sarcopenia of old mice and to elucidate the mechanisms involved. The results suggest that SP might represent a useful dietary supplement to slow the progression of sarcopenia.

## 2. Materials and Methods

### 2.1. SP from Bombyx mori

The SP was supplied by Worldway Co., Ltd. (Sejong, Republic of Korea) and was derived from the cocoons of *Bombyx mori* using a previously described method [27]. Briefly, raw cocoons were acid-hydrolyzed, neutralized, decolorized, filtered, desalted, and freeze-dried to obtain a pale yellow powder. The nutrient composition of the SP was 86.8% protein, 6.78% carbohydrate, 1.79% sodium, 0.94% sugars, and 3.69% other components. The amino acid composition of the SP was determined using high-performance liquid chromatography, as previously described [29]. The most abundant amino acids in the SP were glycine, alanine and serine, accounting for 33.04%, 28.09% and 11.09%, respectively. The SP powder was dissolved in distilled water for oral administration.

### 2.2. Animals and Treatments

The 3–6-month-old and 20–24-month-old female C57BL/6 mice were purchased from the Korea Research Institute of Bioscience and Biotechnology (Daejeon, Republic of Korea). They were cared for according to the standards outlined in the “Guide to the Care and Use of Laboratory Animals” of the National Institutes of Health. Animal experiments were approved by the Institutional Animal Care and Use Committee of CHA University (approval number IACUC210135). The mice were housed at 20 ± 3 °C under a 12 h light–dark cycle and were permitted 1 week to adapt to their surroundings. The mice were randomly assigned to five groups (*n* = 10 per group) as follows: (i) mice aged 3–6 months, as a young control group (YM Ctrl), (ii) mice aged 3–6 months that were administered 750 mg/kg/day SP (YM SP750), (iii) mice aged 20–24 months, as an older control group (OM Ctrl), (iv) mice aged 20–24 months that were administered 100 mg/kg/day SP (OM SP100), and (v) mice aged 20–24 months that were administered 750 mg/kg/day SP (OM SP750). SP solution was orally administered to the mice by gavage daily for 8 weeks. The doses of 100 or 750 mg/kg/day SP were chosen because bioactive effects of SP have previously been demonstrated at these doses [28,29]. At the end of the treatment period, the mice were sacrificed, and tissues were weighed and stored at −80 °C until analyzed.

### 2.3. Reagents

The antibodies used were anti-phospho-forkhead box O3 (p-FoxO3a), FoxO3a phospho-Akt (p-Akt), Akt, phospho-mammalian target of rapamycin (p-mTOR), mTOR (all purchased from Cell Signaling Technology, Danvers, MA, USA), F-box protein (Fbx32, also known as atrogin), muscle ring-finger 1 (MuRF1), myostatin, myoblast determination protein 1 (MyoD), type 2X-myosin heavy chain (MYH1) (all purchased from Abcam, Cambridge, UK), phosphoinositide 3-kinase (PI3K), myogenin, myocyte enhancer factor 2 (MEF-2), superoxide dismutase 1 (SOD1), glutathione peroxidase 1 (GPx1), catalase (all purchased from Santa Cruz Biotechnology, Dallas, TX, USA), fluorescein isothiocyanate (FITC) anti-mouse CD11c, phycoerythrin (PE) anti-mouse CD45, PE anti-mouse CD86, and PerCP/Cyanine5.5 anti-mouse CD206 (all purchased from BioLegend, San Diego, CA, USA).

### 2.4. Measurement of Body Mass and Grip Strength

The body mass and grip strength of the mice were measured 1 day before the start of SP administration and once daily for 8 weeks during the administration period. A Chatillon force measurement system (Columbus Instruments, Columbus, OH, USA) was used to measure all-limb grip strength. Each mouse was placed on the mesh grid, allowing it to grasp the mesh with all four feet, and then its tail was gently pulled three-to-five times to measure the force exerted. The mean grip force of the mouse limbs was recorded.

### 2.5. Histological Analysis

The *quadriceps* and *gastrocnemius* muscles were dissected from the hind limbs of the mice, fixed in 4% paraformaldehyde, embedded in paraffin, and cut into 10 μm sections, which were stained with hematoxylin and eosin (H&E) for histological analysis. The muscle sections were examined under a Nikon E600 microscope (Nikon, Tokyo, Japan) and the cross-sectional areas (CSAs) of the muscle fibers were measured using ImageJ 1.48 software (NIH, Bethesda, MD, USA).

### 2.6. Western Blotting

*Quadriceps* and *gastrocnemius* muscle samples were minced and lysed for 30 min in lysis buffer (iNtRON Biotechnology, Seoul, Republic of Korea) containing protease and phosphatase inhibitors. The protein concentration of each lysate was quantified using a BCA protein assay (Pierce, Rockford, IL, USA). Lysates containing equal amounts of proteins were subjected to SDS-PAGE and then transferred to Immun-Blot PVDF membranes (Bio-Rad, Hercules, CA, USA). After 1 h of blocking in Tris-buffered saline containing 0.05% Tween-20 (TBS-T) and 5% skimmed milk, the membranes were washed in TBS-T and incubated overnight at 4 °C with primary antibody. They were then washed and incubated in TBS-T containing 5% skimmed milk and secondary antibody (peroxidase-conjugated anti-rabbit, anti-mouse, or anti-goat antibody, Bio-Rad, Hercules, CA, USA). Specific protein bands were identified using an EZ-Western Lumi Femto kit (DoGenBio, Seoul, Republic of Korea) and imaged using an LAS-4000 apparatus (GE Healthcare Life Sciences, Marlborough, MA, USA). The relative band intensities were quantified using ImageJ 1.48 software (NIH).

### 2.7. Serum Analysis

Blood samples were collected by cardiac puncture at the time of sacrifice and centrifuged at 800× *g* for 15 min at 4 °C, and the serum samples obtained were stored at −80 °C. The serum concentrations of interleukin (IL)-1β, IL-6, and tumor necrosis factor (TNF)-α were measured using a mouse Th17 magnetic bead panel (Merck Millipore, MA, USA). The serum concentration of insulin-like growth factor (IGF)-1 was measured using an IGF-1 ELISA kit (Thermo Fisher Scientific, Waltham, MA, USA). The serum concentration of glutathione was measured using a Glutathione assay kit (Cayman chemical, Ann Arbor, MI, USA). All these analyses were performed according to the manufacturers’ instructions. Absorbances were measured at appropriate wavelengths using a Luminex 100 analyzer (Luminex, Austin, TX, USA). All measurements were made in triplicate.

### 2.8. Flow Cytometry Analysis

Spleen samples were minced, and single cells were obtained by passing the products through 40 μm strainers. Red blood cells were lysed in an ammonium–chloride–potassium lysis buffer (Lonza, Basel, Switzerland), and after washing, the cells were incubated for 30 min on ice with specific antibodies to identify M1 (CD45^+^CD206^−^CD86^+^) and M2a/M2c (CD45^+^CD206^+^CD86^−^) macrophages.

*Quadriceps* and *gastrocnemius* muscle samples were diced and incubated in collagenase and Dispase II (Sigma-Aldrich, St. Louis, MO, USA) for 1 h at 37 °C. The tissues were then dissociated by passing them through 40 μm strainers and incubated for 30 min on ice with specific antibodies to identify M1 (CD45^+^CD11c^+^CD206^−^) and M2 (CD45^+^CD11c^−^CD206^+^) macrophages.

Flow cytometry was performed using a CytoFlex flow cytometer (Beckman Coulter, Brea, CA, USA) and the data were assessed using FlowJo v10 software (Ashland, OR, USA).

### 2.9. Statistical Analysis

Data are expressed as the mean ± standard error of the mean (SEM), and comparisons were made using one-way analysis of variance (ANOVA), followed by Tukey’s test. *p* < 0.05 was considered to represent statistical significance.

## 3. Results

### 3.1. SP Increases the Strength, Mass, and Size of Muscles of Old Mice without Affecting Body Weight

To evaluate the effects of SP on the age-related loss of muscle strength and mass of mice, young (3–6 months old) and old (20–24 months old) mice were orally administered SP daily for 8 weeks. Their body weight and grip strength were measured weekly, and at the end of the study period, the mice were sacrificed for further analysis of their tissues (Figure 1A). The body weights of neither the young nor the old mice were affected by the SP administration (Figure 1B). However, the grip strength 1 day before the start of the administration was significantly lower in the old group than in the young group. The grip strength of the OM Ctrl group gradually decreased during the 8 weeks of the study, indicating progression of the sarcopenia. However, SP administration improved the grip strength of mice in the OM SP100 and OM SP750 groups over these 8 weeks (Figure 1C). After sacrificing the mice, the *quadriceps* and *gastrocnemius* muscles were excised from their hind limbs and weighed, and the masses normalized to the last-measured body mass were recorded. The masses of the *quadriceps* and *gastrocnemius* muscles of the OM Ctrl group were considerably lower than those of the YM Ctrl and YM SP750 groups, but were dose-dependently increased by SP administration (Figure 1D). Photographs also reveal that muscles removed from the OM Ctrl group were smaller than those removed from the YM Ctrl and YM SP750 groups, but those removed from the OM SP-treated groups were larger (Figure 1E). Taken together, these data indicate that 8 weeks of SP administration ameliorates the sarcopenia of old mice by increasing their muscle strength and mass, without affecting their body weight.

### 3.2. SP Increases Muscle Fiber Size and Reduces the Expression of FoxO3a in Old Mice

We next determined the effect of SP on muscle fiber size. Consistent with the results of grip strength testing, histological analysis of the *quadriceps* and *gastrocnemius* muscles showed that the muscle fibers were, on average, smaller in the OM Ctrl group than in the YM Ctrl group, and that SP administration significantly increased their sizes (Figure 2A,B). The distribution of muscle fiber sizes in each group is shown in the Appendix A.

To elucidate the molecular mechanism by which SP ameliorates sarcopenia, the expression of FoxO3a, which is known to induce muscle wasting by upregulating protein degradation [30], was measured. The phosphorylated form, p-FoxO3a, cannot enter the nucleus and remains in the cytoplasm, where it cannot affect the expression of gene encoding components of the ubiquitin-proteasome system [31]. Lower levels of phosphorylation of FoxO3a were identified in both muscles of the OM Ctrl group than in the YM Ctrl group, but these were increased by the administration of SP, especially in the old mice. This suggests that SP prevents FoxO3a from translocating to the nucleus by decreasing its phosphorylation. We also measured the expression of E3 ubiquitin ligases (Fbx32 and MuRF1), which are activated by FoxO3a signaling and cause polyubiquitination of proteins, leading to skeletal muscle loss [8,9,10]. There was higher expression of Fbx32 and MuRF1 in both muscles of the OM Ctrl group than in the YM Ctrl group. However, SP treatment reduced the expression of both proteins (Figure 2C,D), indicating that SP may reduce skeletal muscle loss by suppressing protein ubiquitination in old mice. Moreover, the expression of myostatin, a muscle cell growth inhibitor, followed the same pattern (Figure 2C,D). 

### 3.3. SP Ameliorates the Defects in Akt/mTOR Signaling in Old Mice

Muscle mass is also upregulated through activation of the Akt/mTOR signaling pathway [32]. The serum concentration of IGF-1, which activates the Akt/mTOR signaling pathway and thereby upregulates muscle protein synthesis, tended to be slightly lower in the OM Ctrl group than in the YM Ctrl group, but was increased by the administration of SP (Figure 3A). The protein expression of phosphoinositol 3-kinase (PI3K), which is activated by IGF-1, showed a similar pattern in the *quadriceps* and *gastrocnemius* muscles of the mice. Therefore, we also measured the phosphorylation and expression of Akt and mTOR in both muscles. In both muscles of the OM Ctrl group, the p-Akt/Akt and p-mTOR/mTOR ratios were significantly lower than in the YM Ctrl group, but the ratios were increased by the administration of SP (Figure 3B,C). These results indicate that SP administration at least partially restores the aging-associated defects in the Akt/mTOR signaling pathway, which would tend to increase protein synthesis.

### 3.4. SP Ameliorates the Defects in the Expression of Myogenic Transcription Factors in Old Mice

To determine whether SP administration might also upregulate myogenesis, the expression of myogenic transcription factors was measured in each of the groups of mice. The protein expression of MyoD, MEF-2, and myogenin in the *quadriceps* and *gastrocnemius* muscles of the OM Ctrl group was lower than that in the YM Ctrl group. However, SP administration increased the expression of all these proteins in both muscles of the old mice (Figure 4A,B). Thus, SP ameliorates the age-related reduction in the expression of myogenic transcription factors, which implies that its effects may be, at least in part, mediated through an upregulation of myogenesis.

### 3.5. SP Ameliorates Low-Grade Chronic Inflammation in the Skeletal Muscle of Old Mice

The onset of sarcopenia is strongly associated with a low-grade chronic inflammatory response that accompanies aging [33]. We measured the mass of the spleen and the numbers of splenocytes in each mouse to determine whether SP administration protects against immunosenescence in old mice. The splenic mass of the OM Ctrl group was significantly higher than that of the YM Ctrl group. However, SP administration reduced this to near the mass of the young mice (Figure 5A), which may be explained by SP suppressing inflammation in the old mice, demonstrated by the enlargement of the spleen. The number of splenocytes showed the same trend (Figure 5B).

We also used flow cytometry to determine whether SP administration affects the phenotype of macrophages, which is important for this inflammation. We defined CD206^−^CD86^+^ cells as M1 macrophages and CD206^+^CD86^−^ cells as M2a/M2c macrophages, and measured the proportion of each type in the spleen of each mouse (Figure 5C). There was a significantly higher proportion of M1 macrophages in the OM Ctrl group than in the YM Ctrl group, but this was normalized by the administration of either dose of SP. Conversely, the proportion of M2 macrophages was lower in the OM Ctrl group than in the YM Ctrl group, and this was increased by the administration of either dose of SP (Figure 5D).

To determine whether SP administration also has an immune-modulating effect in muscle, we defined CD45^+^CD206^−^CD11c^+^ cells as M1 macrophages and CD45^+^CD206^+^CD11c^−^ cells as M2 macrophages, and measured their proportions in the *gastrocnemius* muscle (Figure 5E). In the *gastrocnemius* muscle, similar to the spleen, M1 macrophages accounted for a higher proportion of the total in the OM Ctrl group than in the YM Ctrl group, but the ratio was lower in the OM SP100 and OM SP750 groups. In addition, the proportion of M2 macrophages showed the opposite trends to those of M1 macrophages (Figure 5F).

### 3.6. SP Reduces the Circulating Concentrations of Pro-Inflammatory Cytokines and Oxidative Stress in Old Mice

The above data show that SP administration ameliorates the aging-related changes in the macrophage subtype. Therefore, we next measured the serum concentrations of pro-inflammatory cytokines that are secreted by macrophages (IL-1β, IL-6, and TNF-α). The serum concentrations of these three cytokines were higher in the OM Ctrl group than in the YM Ctrl group, but SP administration reduced the concentrations of all three (Figure 6A). Taken together with the above findings, these results suggest that SP reduces the proportion of M1 macrophages in old mice, which reduces the release of pro-inflammatory cytokines and the low-grade chronic inflammation that is associated with sarcopenia.

In addition to low-grade chronic inflammation, the accumulation of reactive oxygen species in aging skeletal muscle because of oxidative stress is another important component of the pathogenesis of sarcopenia [24]. Therefore, we next determined the effect of SP administration on the protein expression of the antioxidant enzymes that prevent the accumulation of H_2_O_2_ in muscle. We found that the expression of SOD1, GPx1, and catalase in both the *gastrocnemius* and *quadriceps* muscles of mice in the OM Ctrl group was lower than that in the YM Ctrl group, confirming the existence of aging-related oxidative stress. However, SP administration significantly increased the expression of these antioxidant enzymes, especially at the higher dose, in both these muscles in old mice, implying that it protects against oxidative stress (Figure 6B,C). Thus, SP administration may also ameliorate sarcopenia by inhibiting inflammation and oxidative stress in muscle.

## 4. Discussion

Interventions that ameliorate sarcopenia in older people are in increasing demand as the global population ages. The mass and strength of muscle peak at around the age of 30, and after the age of 65 the mass and strength of muscles decrease at a rate that is approximately twice as fast as previously [34]. Thus, sarcopenia becomes increasingly serious in older people. Therefore, we investigated the effects of 8 weeks of administration of SP, a candidate anti-sarcopenia agent, in mice of 3–6 months of age, which corresponds to ~30 years of age in humans, and in mice of >20 months of age, which corresponds to >65 years of age in humans [35].

As expected, we found that the old mice had significantly lower muscle strength, mass, and fiber size than the young mice, indicating that the older mice were sarcopenic. However, the administration of SP to the old mice increased their muscle strength, without affecting their body mass, and increased the size and mass of their *quadriceps* and *gastrocnemius* muscles, which are large muscles of the hind limb. In addition, the size of the muscle fibers, which is directly related to muscle strength, was significantly increased by SP administration in both of these muscles in the old mice. These findings imply that SP ameliorates sarcopenia in old mice, and we next wished to elucidate the mechanism involved.

Inhibition of the breakdown of muscle protein is the primary target for the treatment of sarcopenia. To determine whether SP inhibits protein degradation in muscle, we first determined the effect of SP administration on FoxO3a, a major regulator of muscle protein loss, and found that SP induces the phosphorylation of FoxO3a, which prevents it from translocating to the nucleus and functioning as a transcription factor. SP also reduced the expression of MuRF1 and Fbx32, which mediate proteolysis in both muscles of the old mice. These results suggest that SP inhibits sarcopenia, at least in part, by altering the expression of key mediators of muscle protein breakdown.

Next, the effect of SP on mediators of muscle protein synthesis was investigated. We found that SP administration increases the serum concentration of IGF-1 in old mice, contributing to the promotion of protein synthesis in muscle by activating the PI3K/Akt/mTOR signaling pathway. Moreover, Akt inactivates FoxO3a, meaning that protein degradation mediated by FoxO transcription factors is inhibited by Akt [36,37]. The present findings may also imply that the inhibition of FoxO3a activation and of protein degradation in muscle is exerted through greater phosphorylation of Akt. Thus, SP affects both the Akt/FoxO3a and Akt/mTOR signaling pathways, to respectively reduce the breakdown of muscle proteins and upregulate their synthesis.

SP is also positively involved in myogenesis by promoting the expression of myogenic transcription factors in old mice. Additionally, we measured the expression of myosin heavy chain (MYH), a component of myosis involved in skeletal muscle contraction, to determine whether SP could ameliorate the deficit in muscle contraction caused by aging. The muscle loss observed in sarcopenia is closely related to the loss of type 2 muscle fibers [38,39]. Accordingly, the protein expression levels of the type 2X-myosin heavy chain (MYH1) type were measured in QUA and GAS muscles. As a result, the expression of MYH1 in both muscles decreased in the OM Ctrl group compared to the YM Ctrl group, but increased in the OM SP100 and the OM SP750 group (Appendix A). This result supports the fact that SP contributed to inhibiting type 2 muscle fiber loss, a hallmark of age-induced sarcopenia in old mice.

Immune homeostasis, and in particular the anti-inflammatory capabilities of the body, deteriorate with age, such that a state of chronic low-level inflammation is gradually established. This is characterized by high circulating concentrations of pro-inflammatory cytokines, which promote the breakdown of myofibrillar protein and reduce protein synthesis, thereby directly causing muscle wasting [40]. Given that previous studies have shown that SP improves immune function [27,28,41], we wished to determine whether SP would inhibit this low-grade chronic inflammation in old mice. We found that SP ameliorated indices of immunosenescence and low-grade chronic inflammation in the old mice. Specifically, the SP reduced the age-related phenotypic shift of macrophages from the M1 type, which is pro-inflammatory, to the M2 type, which is involved in tissue repair and suppresses excessive inflammatory responses in the old mice.

Aging is also characterized by a reduction in the ability of the body to defend against cellular damage caused by reactive oxygen species in muscle cells, and the oxidative stress associated with this also leads to muscle impairment. Therefore, we hypothesized that SP might also ameliorate sarcopenia by reducing oxidative stress in the muscle of old mice. We found that SP, at least partially, reverses the age-associated decreases in the muscle expression of antioxidant enzymes, which prevent the local accumulation of H_2_O_2_, in the old mice. These findings imply that administration of SP to old mice may also ameliorate sarcopenia by restoring immune homeostasis, reducing low-grade chronic inflammation, and reducing oxidative stress in muscle.

The YM Ctrl group and the YM SP750 group did not show statistically significant differences in almost all analyses. This is interpreted as the well-maintained homeostasis of immune response and muscle protein metabolism in young mice, regardless of SP administration. On the other hand, SP showed the effect of restoring the broken immune homeostasis and inhibiting the muscle breakdown process promoted by aging in old mice. Therefore, we suggest that SP is effective in delaying the progression of sarcopenia in old mice.

## 5. Conclusions

We have shown that SP affects the activation of the Akt/FoxO3a and Akt/mTOR signaling pathways in old mice, thereby inhibiting the breakdown of muscle proteins and upregulating muscle protein synthesis and myogenesis, which are associated with an amelioration of the age-related reductions in muscle strength, mass, and fiber size. In addition, SP has a protective effect against immunosenescence, low-grade chronic inflammation, and oxidative stress in muscle, which are major causes of sarcopenia. Therefore, SP appears to be an effective means of ameliorating sarcopenia and may represent a useful dietary supplement for use in older people to slow the progression of sarcopenia.

## Figures and Tables

**Figure 1 cells-12-02257-f001:**
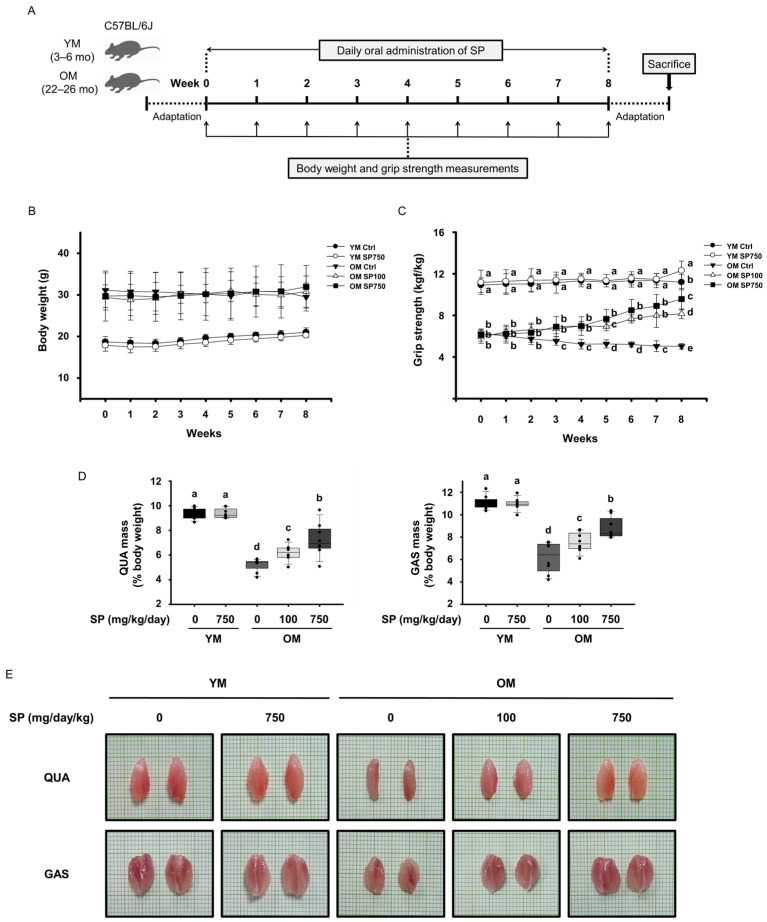
Effects of silk peptide (SP) administration on the strength, mass, and size of the muscles of old mice. (**A**) Experimental protocol: after 1 week of adaptation, young mice (YM) and old mice (OM) were orally administered SP (100 or 750 mg/kg/day) for 8 weeks. The (**B**) body weight and (**C**) grip strength of the mice during the study (mean ± SEM; *n* = 10). (**D**) Masses of the *quadriceps* (QUA, **left**) and *gastrocnemius* (GAS, **right**) muscles at the end of the study, normalized to body weight (mean ± SEM; *n* = 8). (**E**) Representative images of the QUA (**top**) and GAS (**bottom**) muscles. Ctrl, control. Different letters indicate statistically significant differences (*p* < 0.05; one-way ANOVA, followed by Tukey’s test): a > b > c > d > e.

**Figure 2 cells-12-02257-f002:**
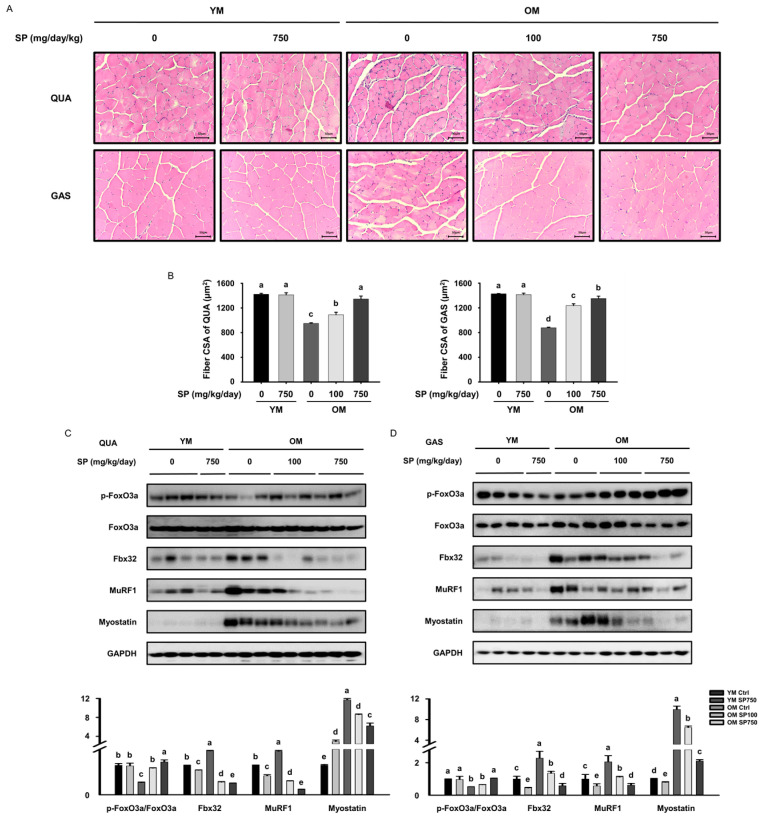
Effects of silk peptide (SP) administration on muscle fiber size and FoxO3a expression in old mice. (**A**) Hematoxylin and eosin-stained transverse sections of the QUA (**top**) and GAS (**bottom**) muscles. (**B**) Cross-sectional areas (CSAs) of the muscle fibers (mean ± SEM; *n* = 4). Expression of proteins mediating muscle loss in the (**C**) QUA and (**D**) GAS muscles, determined using Western blot analysis. GAPDH was used as the loading control for Fbx32, MuRF1, and myostatin, and the phosphorylation of FoxO3a is normalized to the total expression of the protein. Different letters indicate statistically significant differences (*p* < 0.05; one-way ANOVA, followed by Tukey’ test): a > b > c > d > e.

**Figure 3 cells-12-02257-f003:**
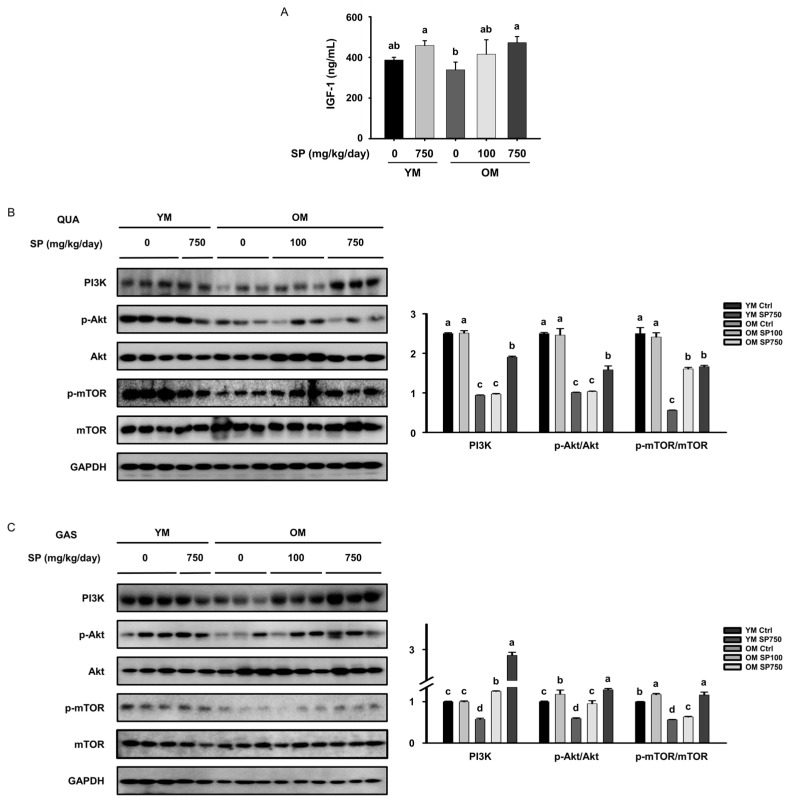
Effect of silk protein (SP) administration on the Akt/mTOR signaling pathway in old mice. (**A**) Serum concentration of IGF-1 (mean ± SEM; *n* = 4). Expression levels of intermediates in the Akt/mTOR pathway, measured using Western blot analysis, in the (**B**) QUA and (**C**) GAS muscles GAPDH was used as a loading control and the phosphorylation of the intermediates was normalized to the total expression of the relevant proteins. Different letters indicate statistically significant differences (*p* < 0.05; one-way ANOVA, followed by Tukey’s test): a > b > c > d.

**Figure 4 cells-12-02257-f004:**
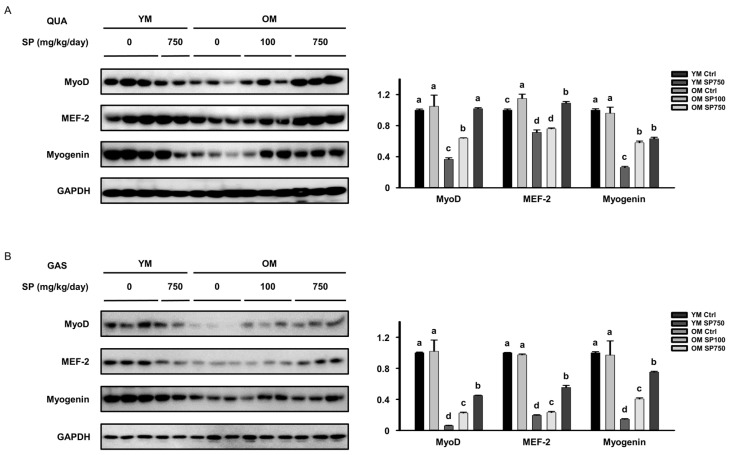
Effect of silk protein (SP) administration on the expression of myogenic transcription factors in old mice. Protein expression of myogenic transcription factors, measured using Western blot analysis, in the (**A**) QUA and (**B**) GAS muscles. Different letters indicate statistically significant differences (*p* < 0.05; one-way ANOVA, followed by Tukey’s test): a > b > c > d.

**Figure 5 cells-12-02257-f005:**
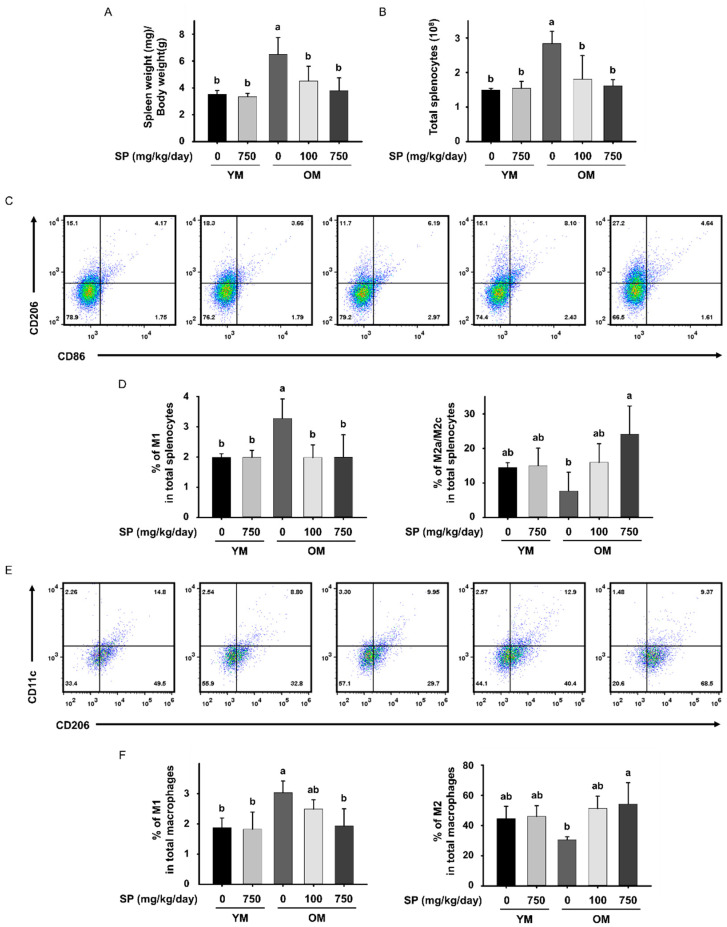
Effects of silk protein (SP) administration on the macrophage subtype and inflammation in old mice. (**A**) Splenic mass, normalized to the body mass of the mice (mean ± SEM; *n* = 8). (**B**) Total number of splenocytes (mean ± SEM; *n* = 3). (**C**) Representative dot plots of CD206^−^CD86^+^ (M1 macrophages) and CD206^+^CD86^−^ cells (M2a/M2c macrophages) in the spleen. (**D**) Bar graph showing the mean percentages of M1 and M2a/M2c macrophages among the splenocytes as a whole (mean ± SEM; *n* = 4). (**E**) Representative dot plots of CD206^−^CD11c^+^ cells (M1 macrophages) and CD206^+^CD11c^−^ cells (M2 macrophages) in the *gastrocnemius* muscle. (**F**) Bar graph showing the mean percentages of M1 and M2 macrophages among all the CD45^+^ cells in muscle (mean ± SEM; *n* = 4). Different letters indicate statistically significant differences (*p* < 0.05; one-way ANOVA, followed by Tukey’s test): a > b.

**Figure 6 cells-12-02257-f006:**
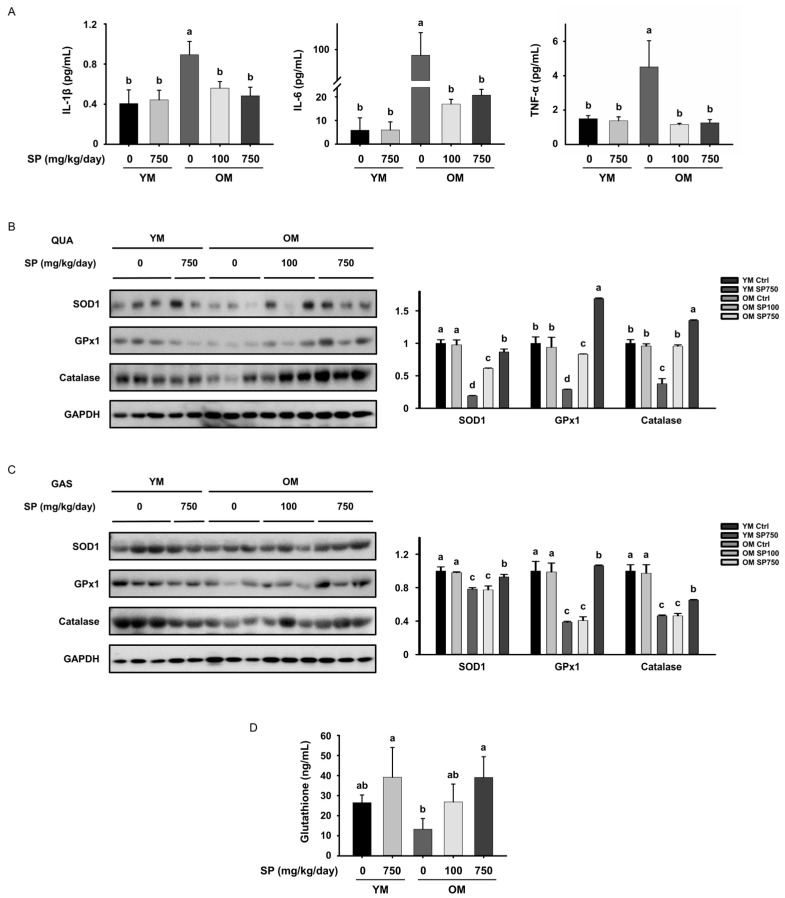
Effect of silk protein (SP) administration on the circulating concentrations of pro-inflammatory cytokines and the expression of antioxidant molecules in old mice. (**A**) Serum concentrations of pro-inflammatory cytokines (IL-1β, IL-6, and TNF-α) (mean ± SEM; *n* = 5). Protein expression of antioxidant enzymes, measured using Western blot analysis, in the (**B**) QUA and (**C**) GAS muscles. GAPDH was used as a loading control. (**D**) Serum concentration of glutathione (mean ± SEM; *n* = 5). Different letters indicate statistically significant differences (*p* < 0.05; one-way ANOVA, followed by Tukey’s test): a > b > c > d.

## Data Availability

All data generated and analyzed during this study are included in this article.

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
