# Peer review of "Silk Peptide Ameliorates Sarcopenia through the Regulation of Akt/mTOR/FoxO3a Signaling Pathways and the Inhibition of Low-Grade Chronic Inflammation in Aged Mice"

_cells, 2023, doi:10.3390/cells12182257_

Round 1

Reviewer 1 Report

Please change all the bar graphs in the figures to show all the individual data points. Thank you.

Author Response

We sincerely appreciate your efforts in reviewing our manuscript.

Based on your comment, we considered changing all bar graphs to individual data points, but there were some difficulties. Since the number of mice per group used in each analysis was not large, it seemed awkward to represent them as individual data points. In addition, we believed that using bar graph with mean ± standard error of the mean (SEM), rather than showing individual data, would provide a clearer way to identify differences or trends between groups. Bar graph effectively visualizes the analysis results and helps to focus on the key features of the data form several groups. In conclusion, we made the decision to retain the bar graphs. We sincerely apologize for not being able to implement the changes you suggested, but we hope for your understanding regarding our perspective. Thank you.

Reviewer 2 Report

I read with great interest the paper entitled: “Silk Peptide Ameliorates Sarcopenia through the Regulation of 2 Akt/mTOR/FoxO3a Signaling Pathways and the Inhibition of 3 Low-Grade Chronic Inflammation in 22-26-Month-Old Mice” from Hyun-Ji Oh and colleagues. This is a very well written manuscript. The introduction is concise and adequate.

For clinical implications of the study, I would have some comments and suggestions:

1. The protein mechanistic explored by the authors is perfect. The authors demonstrated that muscle growth is controlled at the translational level, through the stimulation of protein synthesis, and at the transcriptional level, through the activation of ribosomal RNAs and muscle-specific genes. They also found mTORC1 has a central role in the regulation of both protein synthesis and ribosomal biogenesis. In addition, they found several transcription factors and co-activators, including MEF2, promoted the growth of the myofibers.

2. However, the mechanistic implications on muscle fiber phenotype could have been further explored, considering the practical implications for older people and patients. Histologically, the striated muscle hypertrophy is characterized by large, hypertrophic fibers adjacent to longitudinally split muscle fibers These fibers are often abnormal and can contain one or more internal nuclei, undergo longitudinal, and/or exhibit bizarre cytoarchitectural alterations, such as clefting and whorling. In these instances, ndividual hypertrophic myofibers are often interspersed among atrophic ones. I cannot see this pattern in the Fig 2A, therefore I suggest to the authors improve the resolution of the panel, eventually through high magnifications. In addition, Silk Peptide mice may result from an increase of the number of muscle fibers and probable fiber branching. Did the authors measure the number of fibers? If so, I suggest to the authors include more pictures in the panel of Fig.2A.

3. I also suggest to the authors improve the microscopic description of the muscle. Please, use the following basic topics in the histologic examination of mice fibers.

Organization of skeletal muscle:

Muscle fiber enwrapped by endomysium, loose connective tissue with reticulin and collagen fibers. Fasciculi enwrapped by perimysium, loose connective tissue with elastin and collagen fibers.

Muscle fiber: Elongated cylindrical, unbranched, multinucleated contractile unit formed by fusion of single cells resulting in multinucleated syncytia.

Size: diameter in micrometers; length: in cm.

Flattened nuclei are pushed to periphery of the muscle fiber (just underneath the sarcolemma). Sarcoplasm is filled with myofibrils, a protein structure composed of contractile proteins and organized in a repeating pattern resulting in the light microscopic and electron microscopic appearance of cross striation. Myofibrils composed of thin (actin) and thick (myosin) filaments.

4. Histochemistry stains should be performed to better characterize biochemically the 2 types of fibers:

Type I (aerobic, slow, red):

Small cross section, abundant mitochondria; large amount of myoglobin (oxygen storage)

Type II (anaerobic, fast, white): intense, short, rapid but sporadic movements, large cross section, few mitochondria, little myoglobin, rich in glycogen and glycolytic enzymes.

Masson trichrome: epi, peri and endomysium

Iron hematoxylin: cross striation

PAS - glycogen content:

Type 1 / red fibers: lighter

Type 2 / white fibers: darker

Histochemical stains (to reveal the metabolic properties of red and white fibers):

Succinate dehydrogenase - Kreb cycle enzyme:

Type 1 / red fibers: darker

Type 2 / white fibers: lighter

 5. Electron microscopy, as the gold standard in muscle pathology, may also improve the association with the mechanistic through the analysis of structural components:

I (isotropic) band: thin filaments only

A (anisotropic) band: overlapping thin and thick filaments

H band: thick filaments only

Z line: divides center of I band; serves as attachment site for the sarcomere, the repeating individual unit of the muscle fiber

T tubule: Tubular plasma membrane system / extension at the level of A and I band junction surrounding the myofibril.

Sarcoplasmic reticulum and mitochondria: Smooth endoplasmic reticulum derived membrane system located between the T tubules with sac-like dilatation called terminal cisternae.

Minor revision

Author Response

*Please see the attachment*

Dear Reviewer,

We greatly appreciate your effort in providing us with the opportunity to improve our manuscript through your valuable comments. We have responded step by step to your comments, and we have strived to incorporate your suggestions to the best of our ability. Corrections and additions to the manuscript have been highlighted in red.

Once again, we express our gratitude for providing us with the chance to strengthen our manuscript with your invaluable comments. We hope that the revised manuscript is suitable for publication in Cells.

Yours sincerely,

Professor Boo-Yong Lee

Comment 1. The protein mechanistic explored by the authors is perfect. The authors demonstrated that muscle growth is controlled at the translational level, through the stimulation of protein synthesis, and at the transcriptional level, through the activation of ribosomal RNAs and muscle-specific genes. They also found mTORC1 has a central role in the regulation of both protein synthesis and ribosomal biogenesis. In addition, they found several transcription factors and co-activators, including MEF2, promoted the growth of the myofibers.

Comment 2. However, the mechanistic implications on muscle fiber phenotype could have been further explored, considering the practical implications for older people and patients. Histologically, the striated muscle hypertrophy is characterized by large, hypertrophic fibers adjacent to longitudinally split muscle fibers These fibers are often abnormal and can contain one or more internal nuclei, undergo longitudinal, and/or exhibit bizarre cytoarchitectural alterations, such as clefting and whorling. In these instances, ndividual hypertrophic myofibers are often interspersed among atrophic ones. I cannot see this pattern in the Fig 2A, therefore I suggest to the authors improve the resolution of the panel, eventually through high magnifications. In addition, Silk Peptide mice may result from an increase of the number of muscle fibers and probable fiber branching. Did the authors measure the number of fibers? If so, I suggest to the authors include more pictures in the panel of Fig.2A.

Response 1&2:

Thanks for your in-depth consideration. We improved the resolution of Figure 2A according to your comments, but we could not observe the bizarre cytoarchitectural alterations seen in muscle hypertrophy that you mentioned. We thought that the reason for this is that we focused on the inhibitory effect of SP on age-induced sarcopenia, not on muscle hypertrophy. We demonstrated the effect of delaying the progression of sarcopenia by steadily administering SP to old mice that are naturally losing muscle due to aging. This is different from the range of application that SP just shows a sudden muscle hypertrophy effect. Thus, rather than observing the immediate hypertrophy of individual myofibers, we aimed to observe the trend of overall muscle fiber size change in the old mice group that was consistently administered with SP. And we couldn’t measure the number of muscle fibers. Instead, to present the distribution according to the muscle CSA range in each group in more detail, we provided additional graph in Supplementary data1 (Figure 1S).

Comment 3. I also suggest to the authors improve the microscopic description of the muscle. Please, use the following basic topics in the histologic examination of mice fibers.

Organization of skeletal muscle:

Muscle fiber enwrapped by endomysium, loose connective tissue with reticulin and collagen fibers. Fasciculi enwrapped by perimysium, loose connective tissue with elastin and collagen fibers.

Muscle fiber: Elongated cylindrical, unbranched, multinucleated contractile unit formed by fusion of single cells resulting in multinucleated syncytia.

Size: diameter in micrometers; length: in cm.

Flattened nuclei are pushed to periphery of the muscle fiber (just underneath the 456sarcolemma). Sarcoplasm is filled with myofibrils, a protein structure composed of contractile proteins and organized in a repeating pattern resulting in the light microscopic and electron microscopic appearance of cross striation. Myofibrils composed of thin (actin) and thick (myosin) filaments.

Comment 4. Histochemistry stains should be performed to better characterize biochemically the 2 types of fibers:

Type I (aerobic, slow, red):

Small cross section, abundant mitochondria; large amount of myoglobin (oxygen storage)

Type II (anaerobic, fast, white): intense, short, rapid but sporadic movements, large cross section, few mitochondria, little myoglobin, rich in glycogen and glycolytic enzymes.

Masson trichrome: epi, peri and endomysium

Iron hematoxylin: cross striation

PAS - glycogen content:

Type 1 / red fibers: lighter

Type 2 / white fibers: darker

Histochemical stains (to reveal the metabolic properties of red and white fibers):

Succinate dehydrogenase - Kreb cycle enzyme:

Type 1 / red fibers: darker

Type 2 / white fibers: lighter

 Comment 5. Electron microscopy, as the gold standard in muscle pathology, may also improve the association with the mechanistic through the analysis of structural components:

I (isotropic) band: thin filaments only

A (anisotropic) band: overlapping thin and thick filaments

H band: thick filaments only

Z line: divides center of I band; serves as attachment site for the sarcomere, the repeating individual unit of the muscle fiber

T tubule: Tubular plasma membrane system / extension at the level of A and I band junction surrounding the myofibril.

Sarcoplasmic reticulum and mitochondria: Smooth endoplasmic reticulum derived membrane system located between the T tubules with sac-like dilatation called terminal cisternae.

*3&4&5 are all suggestions for additional experiments, so we responded together.

Response 3&4&5:  

We sincerely appreciate your constructive suggestions. Unfortunately, we could not conduct further experiments for the microscopic description of muscles or the analysis of structural components due to financial and technical problems. Please understand our situation. We demonstrated the potential of SP as a functional food ingredient by investigating its inhibitory effects on age-related muscle loss using old mice. According to the guidelines of the Ministry of Food and Durg Safety in Korea (kMFDS), the biomarkers used to assess sarcopenia are specified as follows (Please see the attachment below). We rigorously evaluated the effects of SP in accordance with these guidelines and aim to seek recognition by the kMFDS as a functional ingredient for health functional foods. We presented results that demonstrate improvements in symptoms of sarcopenia, including muscle strength, muscle size, mass, muscle fiber images, CSA, and more. Considering our challenging circumstances that prevent us from conducting additional experiments that you suggested, we hope for your understanding from the perspective of symptom improvement in age-related sarcopenia.

 However, one thing we could do in your suggestions was to investigate the protein expression level of myosin heavy chain (MYH) through Western blotting. MYH is a component of myosin involved in skeletal muscle contraction, and it is known that muscle loss observed in sarcopenia is associated with a decrease in type 2 muscle fibers [1, 2]. Accordingly, the protein expression level of type 2X-myosin heavy chain (MYH1) in QUA and GAS muscles was measured. As a result, in both muscles, the expression of MYH1 decreased in the OM Ctrl group compared to the YM Ctrl group, but increased in the OM SP100 and especially the OM SP750 groups. This suggests that SP contributed to inhibiting type 2 muscle fiber loss, a hallmark of age-induced sarcopenia, in old mice. These result and contents were added in Supplementary data 2 and Discussion section.

Added to the Discussion section:

In line 378-388 “SP is also positively involved in myogenesis by promoting the expression of myo-genic transcription factors in old mice. Additionally, we measured the expression of myosin heavy chain (MYH), a component of myosis involved in skeletal muscle contraction, to determine whether SP could ameliorate the deficit in muscle contraction caused by aging. The muscle loss observed in sarcopenia is closely related to the loss of type 2 muscle fibers [38, 39]. Accordingly, the protein expression levels of the type 2X-myosin heavy chain (MYH1) type were measured in QUA and GAS muscles. As a result, the expression of MYH1 in both muscles decreased in the OM Ctrl group com-pared to the YM Ctrl group, but increased in the OM SP100 and the OM SP750 group (Figure S2). This result supports that SP contributed to inhibiting type 2 muscle fiber loss, a hallmark of age-induced sarcopenia in old mice.”

[1] Narici, M. V.; Maffulli, N., Sarcopenia: characteristics, mechanisms and functional significance. Br Med Bull 2010, 95, 139-59.

[2]  Tanganelli, F.; Meinke, P.; Hofmeister, F.; Jarmusch, S.; Baber, L.; Mehaffey, S.; Hintze, S.; Ferrari, U.; Neuerburg, C.; Kammerlander, C.; Schoser, B.; Drey, M., Type-2 muscle fiber atrophy is associated with sarcopenia in elderly men with hip fracture. Exp Gerontol 2021, 144, 111171.

Reviewer 3 Report

TThe manuscript Silk Peptide Ameliorates Sarcopenia through the Regulation of Akt/mTOR/FoxO3a Signaling Pathways and the Inhibition of 3 Low-Grade Chronic Inflammation in 22-26-Month-Old Mice is well written and presented, but the authors need to improve some aspects

For example, the length of the title is so long. For example, it is unnecessary to write the months of the mice; I think a shorter title could be more "attractive" for the readers.

Main concerns

The authors did not provide any suggestions on how the SP can initiate this process.

The chemical composition of the SP primarily consists of protein components, although it also contains several additional components. The design of the animal treatment lacks appropriate controls, such as protease treatment of the sample and heat inactivation, in order to gain a more accurate understanding of the potentially involved molecules.

Why did the authors choose not to demonstrate or evaluate the YM SP100 group? Is it feasible to observe an alternative outcome distinct from that of YMSP750?

Minor

Lane 222.  The inactive, (phosphrylated?),  form, p-FoxO3a,

Lanes 232-233 indicating that SP may reduce skeletal muscle loss by suppressing protein 232 ubiquitination in old mice.  The authors can measure the ubiquitination pattern in these tissues by WB. To know more precisely the effect of SP on this process 

Lane 244  …is negatively or positively regulated by the activation or suppression of the AKT…

Lanes 322-lanes 332. The authors can measure oxidative damage to the tissues, for example, lipoperoxidation rate, in order to know the effect of SP.

Some sentences should be eliminated from discussion sections because they are repetitive either in the introduction or results; for example, lines 359–362. Please review all of this section.

Author Response

*Please see the attachment."

Dear Reviewer,

We greatly appreciate your effort in providing us with the opportunity to improve our manuscript through your valuable comments. We have responded point-by-point to your comments, and we have strived to incorporate your suggestions to the best of our ability. Corrections and additions to the manuscript have been highlighted in red.

Once again, we express our gratitude for providing us with the chance to strengthen our manuscript with your invaluable comments. We hope that the revised manuscript is suitable for publication in Cells.

Yours sincerely,

Professor Boo-Yong Lee

Comment 1. The manuscript Silk Peptide Ameliorates Sarcopenia through the Regulation of Akt/mTOR/FoxO3a Signaling Pathways and the Inhibition of 3 Low-Grade Chronic Inflammation in 22-26-Month-Old Mice is well written and presented, but the authors need to improve some aspects. For example, the length of the title is so long. For example, it is unnecessary to write the months of the mice; I think a shorter title could be more "attractive" for the readers.

Response 1: Based on your opinion, the title has been modified as follows by deleting the months of the mice. “Silk Peptide Ameliorates Sarcopenia through the Regulation of Akt/mTOR/FoxO3a Signaling Pathways and the Inhibition of Low-Grade Chronic Inflammation in Aged mice”

Main concerns

Comment 2. The authors did not provide any suggestions on how the SP can initiate this process. The chemical composition of the SP primarily consists of protein components, although it also contains several additional components. The design of the animal treatment lacks appropriate controls, such as protease treatment of the sample and heat inactivation, in order to gain a more accurate understanding of the potentially involved molecules.

Response 2: Thanks for your comment and we agree on the need for more appropriate control. However, we could not set up more control groups because of the limited number of old mice and financial problems (extremely high cost of old mice, cost of animal care, etc.). But the important thing is that SP is not a single compound, but a natural material isolated from Bombyx mori, and protein constitutes most of it.  Several in vitro and in vivo studies on the bioactive effect of naturally derived silk fibroin and silk protein have been reported [1-4]. Based on these reported various physiological activity effects of SP, we administered SP to old mice and evaluated whether it was effective against aging-induced sarcopenia.

 Since SP contains dipeptides and tripeptides, including amino acids, dietary SP is a good source of nutrient for intestinal absorption. As mentioned in line 96-97, the most abundant amino acids in the SP were glycine, alanine and serine (See also the HPLC chromatogram for the SP and Free amino acid composition of the protein component of SP attached below). Among them, glycine and serine in the SP have been designated as functional components of silk peptide by the Ministry of Food and Drug Safety in Korea. In addition, based on several studies reported that glycine and serine have a positive effect on muscles, it is assumed that they played an important role in improving sarcopenia in this study [5-8].

[1] Chaturvedi, V.; Naskar, D.; Kinnear, B. F.; Grenik, E.; Dye, D. E.; Grounds, M. D.; Kundu, S. C.; Coombe, D. R., Silk fibroin scaffolds with muscle-like elasticity support in vitro differentiation of human skeletal muscle cells. J Tissue Eng Regen Med 2017, 11, (11), 3178-3192.

[2] Jung, E. Y.; Lee, H. S.; Lee, H. J.; Kim, J. M.; Lee, K. W.; Suh, H. J., Feeding silk protein hydrolysates to C57BL/KsJ-db/db mice improves blood glucose and lipid profiles. Nutr Res 2010, 30, (11), 783-90.

[3] Do, S. G.; Park, J. H.; Nam, H.; Kim, J. B.; Lee, J. Y.; Oh, Y. S.; Suh, J. G., Silk fibroin hydrolysate exerts an anti-diabetic effect by increasing pancreatic beta cell mass in C57BL/KsJ-db/db mice. J Vet Sci 2012, 13, (4), 339-44.

[4] Lee, H. S.; Lee, H. J.; Suh, H. J., Silk protein hydrolysate increases glucose uptake through up-regulation of GLUT 4 and reduces the expression of leptin in 3T3-L1 fibroblast. Nutr Res 2011, 31, (12), 937-43.

[5] Gheller, B. J.; Blum, J. E.; Lim, E. W.; Handzlik, M. K.; Hannah Fong, E. H.; Ko, A. C.; Khanna, S.; Gheller, M. E.; Bender, E. L.; Alexander, M. S.; Stover, P. J.; Field, M. S.; Cosgrove, B. D.; Metallo, C. M.; Thalacker-Mercer, A. E., Extracellular serine and glycine are required for mouse and human skeletal muscle stem and progenitor cell function. Mol Metab 2021, 43, 101106.

[6] Caldow, M. K.; Ham, D. J.; Trieu, J.; Chung, J. D.; Lynch, G. S.; Koopman, R., Glycine Protects Muscle Cells From Wasting in vitro via mTORC1 Signaling. Front Nutr 2019, 6, 172.

[7] Koopman, R.; Caldow, M. K.; Ham, D. J.; Lynch, G. S., Glycine metabolism in skeletal muscle: implications for metabolic homeostasis. Curr Opin Clin Nutr Metab Care 2017, 20, (4), 237-242.

[8] Liu, Y.; Wang, X.; Wu, H.; Chen, S.; Zhu, H.; Zhang, J.; Hou, Y.; Hu, C. A.; Zhang, G., Glycine enhances muscle protein mass associated with maintaining Akt-mTOR-FOXO1 signaling and suppressing TLR4 and NOD2 signaling in piglets challenged with LPS. Am J Physiol Regul Integr Comp Physiol 2016, 311, (2), R365-73.

Comment 3. Why did the authors choose not to demonstrate or evaluate the YM SP100 group? Is it feasible to observe an alternative outcome distinct from that of YMSP750?

Response 3: Firstly, the YM Ctrl group and the YM SP750 group did not show statistically significant differences in nearly all of the analyses. Recognizing the importance of maintaining homeostasis in immune response and muscle protein metabolism, these results are interpreted as the YM group effectively maintaining homeostasis throughout regardless of SP administration. Sarcopenia is characterized by age-related muscle loss. Since our focus was on improving “age-related” sarcopenia, we chose to delve more deeply into discussing the effects of SP on alleviating age-related muscle loss, particularly in old mice, rather than dwelling on the outcomes in the YM group where no significant differences were observed.

 It is in the same context as above that the YM SP100 group was not evaluated. We have already confirmed that the effect of high-dose SP did not cause significant changes in young mice who are not undergoing aging-related muscle loss. And we judged it meaningless to discuss the effect of low-dose SP. Therefore, just like YM SP750, we did not evaluate the YM SP100 group, as it was no expected to significantly differ from the YM control group.

We added the following to the Discussion section:

In line 409-415 “The YM Ctrl group and the YM SP750 group did not show statistically significant differences in almost all analyses. This is interpreted as well-maintained homeostasis of immune response and muscle protein metabolism in young mice, regardless of SP administration. On the other hand, SP showed the effect of restoring the broken immune homeostasis and inhibiting the muscle breakdown process promoted by aging in old mice. Therefore, we suggest that SP is effective in delaying the progression of sarcopenia in old mice.”

Minor

Comment 4. Lane 222.  The inactive, (phosphrylated?),  form, p-FoxO3a,

Response 4: We edited the sentence.

In line 224 “The phosphorylated form, p-FoxO3a, ~”

Comment 5. Lanes 232-233 indicating that SP may reduce skeletal muscle loss by suppressing protein 232 ubiquitination in old mice.  The authors can measure the ubiquitination pattern in these tissues by WB. To know more precisely the effect of SP on this process 

Response 5: Thanks for your suggestion. It is well known that increased activity of MuRF-1 and Fbx32, which are E3 ubiquitin ligases expressed in skeletal muscle, plays a major role in skeletal muscle loss through protein polyubiquitination [9-11]. Although we did not directly measure the ubiquitination pattern in muscles, the references below support our rationale. We also added relevant references and edited the sentence for clarification.

Revised line 230-232 “We also measured the expression of E3 ubiquitin ligases (Fbx32 and MuRF1), which are activated by FoxO3a signaling and cause polyubiquitination of proteins, leading to skeletal muscle loss [8-10].”

[9] Kang, S. H.; Lee, H. A.; Kim, M.; Lee, E.; Sohn, U. D.; Kim, I., Forkhead box O3 plays a role in skeletal muscle atrophy through expression of E3 ubiquitin ligases MuRF-1 and atrogin-1 in Cushing's syndrome. Am J Physiol Endocrinol Metab 2017, 312, (6), E495-E507.

[10] Rom, O.; Reznick, A. Z., The role of E3 ubiquitin-ligases MuRF-1 and MAFbx in loss of skeletal muscle mass. Free Radic Biol Med 2016, 98, 218-230.

[11] Gumucio, J. P.; Mendias, C. L., Atrogin-1, MuRF-1, and sarcopenia. Endocrine 2013, 43, (1), 12-21.

Comment 6. Lane 244  …is negatively or positively regulated by the activation or suppression of the AKT…

Response 6: According to your comment, we edited the sentence for clarification.

Revised line 247 “Muscle mass is also upregulated through activation of the Akt/mTOR signaling pathway [32].”

Comment 7. Lanes 322-lanes 332. The authors can measure oxidative damage to the tissues, for example, lipoperoxidation rate, in order to know the effect of SP.

Response 7: Thanks for your suggestion. Unfortunately, we could not measure lipoperoxidation rate due to financial and technical limitations. Please understand our situation. However, as shown in our results (Fig 6), the significantly lower expression of antioxidant enzymes in the OM Ctrl group compared to the YM Ctrl group proves the existence of oxidative stress in both muscles of old mice. Deficiencies in antioxidant enzymes in muscle causes accumulation of free radicals, leading to cellular damage and ultimately, muscle tissue damage [12-14]. In the OM Ctrl group, where muscle loss is prominent, damage to muscle tissue is expected. But SP increased the expression of antioxidant enzymes in muscles of old mice. This ultimately means that SP suppressed muscle damage by reducing oxidative stress, contributing to the improvement of sarcopenia.

[12] Roy, Z.; Bansal, R.; Siddiqui, L.; Chaudhary, N., Understanding the Role of Free Radicals and Antioxidant Enzymes in Human Diseases. Curr Pharm Biotechnol 2023, 24, (10), 1265-1276.

[13] Chaudhary, P.; Janmeda, P.; Docea, A. O.; Yeskaliyeva, B.; Abdull Razis, A. F.; Modu, B.; Calina, D.; Sharifi-Rad, J., Oxidative stress, free radicals and antioxidants: potential crosstalk in the pathophysiology of human diseases. Front Chem 2023, 11, 1158198.

[14] Kozakowska, M.; Pietraszek-Gremplewicz, K.; Jozkowicz, A.; Dulak, J., The role of oxidative stress in skeletal muscle injury and regeneration: focus on antioxidant enzymes. J Muscle Res Cell Motil 2015, 36, (6), 377-93.

Comment 8. Some sentences should be eliminated from discussion sections because they are repetitive either in the introduction or results; for example, lines 359–362. Please review all of this section.

Response 8: According to your comment, we’ve appropriately modified Discussion section.

Round 2

Reviewer 1 Report

Dear Authors,

It is necessary to show the individual data points in your bar graphs. It is now standard practice for all animal experiments to show the individual data points. You have mentioned that each group had n=10 mice, so why not show the values for each of these 10 mice? Showing just the mean with SEM is hiding all these information from the readers and the scientific community. I would insist on plotting the bar graphs with the individual values using Graph Pad Prism or any similar software. I would recommend a major revision yet again. Thank you.

Author Response

Again, we sincerely appreciate your efforts in reviewing our manuscript.

Response: Thank you for your comment. While we designated 10 mice per group, it’s important to note that not all 10 mice were used in each analysis. We believe you are also aware that it is challenging for every mouse to participate in every analysis. We allocated the appropriate number of animals for each analysis and presented statistical information using bar graphs. Bar graphs that provide mean values, standard errors, and confidence intervals (even without individual data) is very commonly used in animal experiments, including many animal experiments papers published in Cells. Not all papers, which present animal experiment results in bar graphs without individual data, lack reliability or hide their information. As in the previous answer, we decided to keep our representation of bar graphs considering the following advantages: Provides a clearer way to identify differences or trends between groups, effectively visualizes analysis results, helps focus on key features of data form across multiple groups, and provides intuitive interpretation.

However, we have made efforts to address the areas of concern you raised, aiming to better align with your feedback. We indicated the number of animals (‘n’) used in each analysis in each Figure Legend. Thank you for your advice so we could provide more clear information. Once again, we sincerely apologize for not being able to implement the changes you suggested, but we hope for your understanding regarding our perspective. Thank you.

Revised line 211, 213, 242, 263, 308, 309, 311, 314, 342, 344

Reviewer 2 Report

The authors answered all questions raised, justifying their inclusion or not in the new manuscript. I consider the manuscript to be relevant to readers in terms of methodology and mechanics. I therefore endorse your publication. Thank you for the opportunity to review this article.

Minor editing

Author Response

We sincerely appreciate your efforts in reviewing our manuscript and your support for its publication. Thank you.